# Towards Novel Treatments for Schizophrenia: Molecular and Behavioural Signatures of the Psychotropic Agent SEP-363856

**DOI:** 10.3390/ijms22084119

**Published:** 2021-04-16

**Authors:** Veronica Begni, Alice Sanson, Alessia Luoni, Federica Sensini, Ben Grayson, Syeda Munni, Joanna C. Neill, Marco A. Riva

**Affiliations:** 1Laboratory of Psychopharmacology and Molecular Psychiatry, Department of Pharmacological and Biomolecular Sciences, University of Milan, Via Balzaretti 9, 20133 Milan, Italy; veronica.begni@unimi.it (V.B.); alice.sanson@unimi.it (A.S.); alessia.luoni@live.it (A.L.); federica.sensini@gmail.com (F.S.); 2Manchester Academic Health Science Centre, Division of Pharmacy and Optometry, School of Health Sciences, Faculty of Medicine, Biology and Health, University of Manchester, Manchester M13 9PT, UK; ben.grayson@manchester.ac.uk (B.G.); syeda.munni@manchester.ac.uk (S.M.); joanna.neill@manchester.ac.uk (J.C.N.); 3Biological Psychiatry Laboratory, IRCCS Istituto Centro San Giovanni di Dio Fatebenefratelli, Via Pilastroni, 4, 25125 Brescia, Italy

**Keywords:** schizophrenia, antipsychotic drug, SEP-363856, trace amine-associated receptor, phencyclidine, amphetamine

## Abstract

Schizophrenia is a complex psychopathology whose treatment is still challenging. Given the limitations of existing antipsychotics, there is urgent need for novel drugs with fewer side effects. SEP-363856 (SEP-856) is a novel psychotropic agent currently under phase III clinical investigation for schizophrenia treatment. In this study, we investigated the ability of an acute oral SEP-856 administration to modulate the functional activity of specific brain regions at basal levels and under glutamatergic or dopaminergic-perturbed conditions in adult rats. We found that immediate-early genes (IEGs) expression was strongly upregulated in the prefrontal cortex and, to a less extent, in the ventral hippocampus, suggesting an activation of these regions. Furthermore, SEP-856 was effective in preventing the hyperactivity induced by an acute injection of phencyclidine (PCP), but not of *d*-amphetamine (AMPH). The compound effectively normalized the PCP-induced increase in IEGs expression in the PFC at all doses tested, whereas only the highest dose determined the major modulations on AMPH-induced changes. Lastly, SEP-856 acute administration corrected the cognitive deficits produced by subchronic PCP administration. Taken together, our data provide further insights on SEP-856, suggesting that modulation of the PFC may represent an important mechanism for the functional and behavioural activity of this novel compound.

## 1. Introduction

Schizophrenia is a chronic and severe psychiatric disorder that affects approximately 20 million people worldwide [1]. It is characterized by a variety of clinical manifestations, of three main types: positive symptoms (hallucinations and thought alterations), negative symptoms (impaired social interaction and avolition) and cognitive deficits (attention and memory impairments) [2]. Antipsychotic drugs (APDs) represent the mainstay for the treatment of schizophrenia, acting through the modulation of dopamine D_2_ and serotonin 5-HT_2 A_ receptors, although second generation APDs are considered multireceptor modulators [3]. While these drugs are effective in alleviating positive symptoms of schizophrenia, they show variable responses on negative symptoms and cognitive dysfunction. For this reason, there is an urgent need to develop novel compounds to provide a more effective management of the disorder [4,5]. To this end, SEP-363856 [(S)-1-(4,7-dihydro-5H-thieno[2,3-c]pyran-7-yl)-*n*-methylmethanamine hydrochloride] is a compound with a novel mechanism that demonstrated antipsychotic activity in a phase II study [6] and is currently under phase III of clinical development. Although its mechanism of action has not been completely elucidated, SEP-363856 (SEP-856) mainly acts as agonist at serotonin 5-HT_1 A_ and trace amine-associated (TAAR1) receptors [7]. 5-HT_1 A_ receptors are mostly expressed as presynaptic autoreceptors on serotoninergic neurons in the dorsal raphe nucleus (DRN) and as postsynaptic heteroreceptors on nonserotoninergic neurons in various brain regions, including hippocampus, prefrontal cortex, amygdala and hypothalamus [8]. On the other hand, TAAR1, a member of the TAAR family, is a G protein-coupled receptor, mostly expressed in the ventral tegmental area (VTA) and DRN [9,10]. TAAR1 modulates monoamine transmission, as it can interact and regulate dopaminergic and serotoninergic signalling by inhibiting D_2_ while activating 5-HT_1 A_ receptors [11]. Considering its modulatory functions, this receptor is becoming a promising target for pharmacological intervention. Accordingly, a previous preclinical characterization of SEP-856 showed its effectiveness in alleviating behaviours reflecting both positive and negative schizophrenia symptomatology, suggesting a broad spectrum of therapeutic efficacy [7]. Moreover, SEP-856 proved to be effective in reducing the increase of dopamine synthesis that followed subchronic ketamine administration [12].

Since one key aspect of APDs is their ability to modulate the activity of specific brain circuits and structures, the aim of the present study was to investigate in more detail such mechanisms in response to SEP-856 administration. To this end, we first conducted an explorative experiment in male rats, investigating the effects of an acute administration of SEP-856 on the expression of immediate-early genes (IEGs) in different brain regions such as prefrontal cortex, dorsal and ventral hippocampus and striatum, which play a pivotal role in the pathophysiology of schizophrenia. To the best of our knowledge, although the selective agonism of TAAR1 and 5-HT_1 A_, in addition to the lack of D_2_/5-HT_2 A_-mediated efficacy has been proved [7], no study so far has investigated the ability of SEP-856 to modulate specific brain regions in healthy animals under basal conditions.

Next, considering the limited ability of molecules lacking D_2_ receptor affinity in alleviating schizophrenia symptomatology [4], we investigated whether an acute administration of SEP-856 could ameliorate the behavioural and molecular alterations induced by psychotomimetic drugs that may reproduce key functional alterations observed in schizophrenia. To this purpose, we tested the ability of SEP-856 in modulating the behavioural and functional responses to an acute administration of either phencyclidine (PCP) or *d*-amphetamine (AMPH) in female rats, which are more sensitive to psychostimulants [13,14]. PCP is a noncompetitive antagonist at the glutamatergic *n*-methyl-D-aspartate (NMDA) receptor that may mimic a hypoglutamatergic state thought to be relevant for different schizophrenia-like symptoms [15]. Alternatively, AMPH is able to enhance dopamine release, thus leading to a hyperdopaminergic state that underlies the positive symptoms of schizophrenia [16]. Last, to evaluate the therapeutic potential of SEP-856 in schizophrenia, we investigated whether acute administration of SEP-856 could improve the cognitive deficits originating from a subchronic PCP regimen, a valuable tool to induce a schizophrenia-like phenotype in rodents, particularly cognitive dysfunctions of relevance to the illness (see [17,18] for reviews).

## 2. Results

### 2.1. Acute SEP-856 Administration Up-Regulates the Expression of Activity-Regulated Genes with Anatomical Selectivity

As a first step, we investigated the ability of SEP-856 to modulate the activity of different brain regions after acute administration. First we measured the expression of the activity-regulated cytoskeleton associated protein (*Arc*), an immediate-early gene involved in neuroplasticity, determining memory formation and sustaining cognitive processes [19,20]. As shown in Figure 1A, *Arc* mRNA levels were significantly modulated by SEP-856 treatment in the PFC (F(3,24) = 10.193, *p* = 0.000162) and ventral-HIP (F(3,24) = 4.964, *p* = 0.008). In detail, SEP-856 treatment strongly upregulated the expression of *Arc* in the PFC at all doses tested (1 mg/kg, *p* < 0.001; 3 and 10 mg/kg, *p* < 0.01). Conversely, in the ventral-HIP, *Arc* mRNA levels were significantly upregulated only by the lowest dose (1 mg/kg, *p* < 0.05). In contrast, acute SEP-856 treatment was not able to alter *Arc* mRNA levels within the striatum (F(3,33) = 0.815, *p* = 0.495) or dorsal-HIP (F(3,26) = 0.295, *p* = 0.829).

A similar pattern of modulation was observed for the cellular oncogene C-Fos (*c-Fos*), a regulatory transcription factor important for learning and memory [21] (Figure 1B). Indeed, its mRNA levels were significantly increased after acute SEP-856 treatment within the PFC (F(3,24) = 14.471, *p* = 0.000014) and the ventral-HIP (F(3,26) = 4.123, *p* = 0.016). Similar to *Arc* expression, all doses of the compound were able to produce a significant upregulation of *c-Fos* expression within the PFC (1 and 3 mg/kg, *p* < 0.001; 10 mg/kg, *p* < 0.01). In the ventral-HIP, *c-Fos* mRNA levels were significantly induced by the compound at the doses of 3 and 10 mg/kg (*p* < 0.05). Conversely, and as for *Arc*, the acute administration of SEP-856 did not produce any significant change within the striatum (F(3,36) = 1.329, *p* = 0.280) or the dorsal-HIP (F(3,31) = 1.007, *p* = 0.403).

Next, we analysed the expression of the zinc finger binding protein clone 268 (*Zif268/Egr1*), a transcription factor involved in several neuronal plasticity processes [22]. As shown in Figure 1C, *Zif268/Egr1* mRNA levels were also significantly upregulated by acute SEP-856 treatment within the PFC (F(3,34) = 6.927, *p* = 0.001). Specifically, this effect was significant for the lowest and highest doses of the compound (1 and 10 mg/kg, *p* < 0.01), while the increase observed in animals treated with SEP-856 at 3 mg/kg did not reach statistical significance (*p* = 0.231). Furthermore, the acute administration of SEP-856 did not produce significant changes of the *Zif268/Egr1* mRNA levels in the ventral (F(3,25) = 2.173, *p* = 0.116) and dorsal (F(3,33) = 0.421, *p* = 0.739) hippocampus, or in the striatum (F(3,33) = 1.694, *p* = 0.187).

Lastly, we investigated neuronal PAS domain protein (*Npas4*), a neuronal transcription factor that regulates the excitatory/inhibitory balance [23,24]. As shown in Figure 1D, we found that SEP-856 was able to induce its mRNA levels only within the PFC (F(3,35) = 10.226, *p* = 0.000056). In detail, the acute administration of the compound at the dose of 1 and 3 mg/kg produced a significant increase of *Npas4* mRNA levels (1 mg/kg, *p* < 0.001; 3 mg/kg, *p* < 0.05). Conversely, in line with the results for other IEGs expression, *Npas4* mRNA levels were not modulated by acute SEP-3856 treatment within either the hippocampal subregions (Ventral-HIP: F(3,30) = 1.442, *p* = 0.250; Dorsal-HIP: F(3,34) = 0.692, *p* = 0.563) or within the striatum (F(3,35) = 2.085, *p* = 0.120). 

### 2.2. Modulation of PCP-Induced Hyperlocomotion after Acute SEP-856 Administration

To investigate the antipsychotic-like activity of SEP-856, we measured its ability to modulate the total locomotor activity in animals treated with PCP, mimicking a pathological condition. Figure 2A shows a comprehensive analysis of locomotor activity over the total 150 min evaluation. Repeated measures ANOVA showed a significant main effect of time and treatment, but only a slight tendency towards significance for time*treatment interaction (main effect of time: F(7.860,337.984) = 56.832, *p* = 1.88 × e^−57^, ηp^2^ = 0.569; main effect of treatment: F(4,43) = 5.973, *p* = 0.0006, ηp^2^ = 0.357; time*treatment: F(31.440,337.984) = 1.396, *p* = 0.082, ηp^2^ = 0.115). Indeed, PCP administration induced a significant immediate hyperactive state (*p* < 0.05), maintained throughout the behavioural evaluation, that was attenuated by the pretreatment with SEP-856 at any dose (*p* < 0.01 for all doses vs. PCP). 

Based on that, we further analysed the behavioural data focusing on the total movements before and after PCP administration individually. As Figure 2B shows, statistical analysis of the total movements during the first 60 min of the test revealed a significant effect of drug treatment (F(3,44) = 7.067, *p* = 0.0006). Indeed, as confirmed by post hoc comparisons, the treatment with SEP-856 at all doses significantly reduced total activity within the first part of the test, i.e., before administration of PCP (*p* < 0.05 for 1 mg/kg, *p* < 0.01 for 3 and 10 mg/kg compared with vehicle control). Similarly, we observed a significant effect of drug treatment when analysing the second part of the test, post-PCP administration (65–150 min; F(4,43) = 4.895, *p* = 0.002). Indeed, PCP administration induced a robust and significant increase in total activity compared to vehicle control (*p* < 0.01), an effect which SEP-856 significantly attenuated (*p* < 0.05 for 1 mg/kg, *p* < 0.01 for 3 and 10 mg/kg, compared with PCP alone) (Figure 2C). 

### 2.3. Analysis of IEGs Expression in Rat Prefrontal Cortex Following PCP Administration: Modulation by Acute Pretreatment with SEP-856

Based on the significant changes observed in the expression of IEGs within the PFC after acute SEP-856 administration in the explorative experiment, and significant attenuation of PCP-induced hyperlocomotion, we decided to investigate if pretreatment with the compound could modulate the effects of an acute PCP injection on the expression of IEGs in this specific brain region.

The statistical analysis of *Arc* gene expression showed a significant effect of drug treatment (F(4,36) = 3.152, *p =* 0.025). Indeed, as shown in Figure 3A, acute PCP administration produced an increase of *Arc* mRNA levels, although this effect did not reach statistical significance (+52%, *p* = 0.072). This mild activation of *Arc* expression was prevented by the pretreatment with SEP-856. Specifically, the administration of the compound at 3 mg/kg and 10 mg/kg doses appeared to be more effective in preventing the PCP-induced changes, as confirmed by post hoc comparison (*p* < 0.05 and *p* = 0.066 vs. PCP-treated animals, with 3 and 10 mg/kg respectively).

When investigating *c-Fos* expression (Figure 3B), univariate ANOVA revealed a significant modulation by drug treatment (F(4,33) = 3.279, *p =* 0.023). Indeed, acute PCP administration produced a significant increase of *c-Fos* mRNA levels (+60%, *p* < 0.05). While this effect was not counteracted by the lowest dose of SEP-856 (1 mg/kg) (+59%, *p* < 0.05 vs. vehicle treated rats), the doses of 3 and 10 mg/kg were able to mitigate the upregulation of *c-Fos* expression (*p* = 0.057 and *p* > 0.05 vs. vehicle treated rats). However, none of the administered doses of SEP-856 induced a modulation statistically different compared to PCP treatment alone.

Lastly, drug treatment did not produce any significant modulation of *Zif268/Egr1* (Figure 3C; F(4,40) = 2.194, *p* = 0.087) and *Npas4* (Figure 3D; F(4,38) = 1.043, *p =* 0.398) mRNA levels.

Next, in order to establish if the ability of SEP-856 in preventing PCP-induced hyperactivity could be associated with the molecular changes of IEGs expression, we calculated the Pearson product-moment correlation coefficient (*r*) between the total movements measured in the second part of the test (65–150 min) and mRNA levels of *Arc*, *c-Fos*, *Zif268/Egr1* and *Npas4*. As depicted in Figure 4, we found a significant positive correlation between the total locomotor activity and *Arc* mRNA levels (Figure 4A; *r* = 0.352, *p* < 0.05). Conversely, no significant correlations were found between the behavioural performance and the expression of other IEGs.

### 2.4. Modulation of AMPH-Induced Hyperlocomotion after Acute SEP-856 Administration

To further characterize SEP-856 activity, we measured the total locomotor activity following an injection of AMPH, which mimics a hyperdopaminergic state, in animals pretreated with SEP-856 or vehicle.

Repeated measures of ANOVA over the 150 min evaluation showed a significant main effect of time, treatment and time*treatment interaction (main effect of time: F(7.250, 311.750) = 33.438, *p* = 2.04 × e^−35^, ηp^2^ = 0.437; main effect of treatment: F(4,43) = 3.378, *p* = 0.017, ηp^2^ = 0.239; time*treatment: F(29.000, 311.750) = 4.617, *p* = 2.3 × e^−12^, ηp^2^ = 0.300). Indeed, as shown in Figure 5A, AMPH produced a strong increase of the locomotor activity (*p* < 0.05) that was not prevented by pretreatment with any doses of SEP-856 (1 mg/kg: *p* < 0.05; 10 mg/kg: *p* = 0.066 vs. vehicle-treated rats). Moreover, when analysing the total counts during the first 60 min of the test (Figure 5B), univariate ANOVA showed an almost significant effect of drug treatment (F(3,44) = 2.643, *p* = 0.061). Focusing on the second part of the test, we found a significant effect of drug treatment (F(4,43) = 5.494, *p* = 0.001), as confirmed by post hoc comparisons. Indeed, AMPH administration produced a significant increase in total locomotor activity compared to vehicle-treated rats (*p* < 0.01). However, the acute pretreatment with SEP-856 did not prevent this effect at any dose (*p* < 0.05 for 3 mg/kg; *p* < 0.01 for 1 and 10 mg/kg vs. vehicle treated rats). Accordingly, behavioural performances of SEP-856 pretreated animals did not significantly differ from AMPH-injected rats (*p* > 0.05 for all doses vs. AMPH) (Figure 5C).

### 2.5. Analysis of IEGs Expression in Rat Prefrontal Cortex Following AMPH Administration: Modulation by Acute Pretreatment with SEP-856 

Although acute SEP-856 administration was not able to prevent the hyperlocomotion produced by AMPH, we decided to examine the pattern of IEGs expression to investigate the potential of SEP-856 to modulate PFC activation under perturbed dopaminergic function.

When investigating *Arc* mRNA expression, univariate ANOVA showed a significant effect of drug treatment (F(4,40) = 4.95, *p =* 0.004). Indeed, as shown in Figure 6A, acute AMPH administration produced an increase of *Arc mRNA* levels, although the effect did not reach statistical significance (+48%, *p* = 0.059). The elevation of *Arc* expression was prevented by the acute pretreatment with SEP-856 at all doses, with the main effect at 10 mg/kg (*p* < 0.01 vs. AMPH-treated animals).

Conversely, analysis of the other IEGs did not reveal a significant effect of drug treatment, suggesting that the expression of *c-Fos* (Figure 6B; F(4,41) = 1.644, *p =* 0.182), *Zif268/Egr1* (Figure 6C; F(4,39) = 2.140, *p =* 0.094) *and Npas4* (Figure 6D; F(4,40) = 1.823, *p =* 0.143) was not significantly affected by AMPH administration alone or following pretreatment with any dose of SEP-856. 

### 2.6. Acute SEP-856 Administration Attenuates the Cognitive Deficits Induced by Sub-Chronic PCP Treatment

In order to substantiate the potential effectiveness of SEP-856 on a pathological domain relevant for schizophrenia, we decided to investigate the ability of an acute treatment with the compound on cognitive function in the subchronic PCP rat model using the novel object recognition test (NOR) [25]. Although no significant difference was observed in the exploration times of each object during the acquisition trial of the task, as well as in the total exploration times of each trial (Figure 7A), the subchronic treatment with PCP determined a strong impairment in the retention trial: scPCP-treated animals spent an equal amount of time exploring the familiar and the novel object (Figure 7B). While this deficit persisted in animals treated with SEP-856 at dose 1 mg/kg, we found that rats acutely treated with the highest dose of SEP-856 spent significantly longer time exploring the novel object (*p* < 0.001 novel vs. familiar). As further confirmed by the analysis of the discrimination index (DI; ANOVA: F(3,24) = 12.293, *p* = 0.000045), scPCP strongly reduced cognitive performance (Figure 7C, *p* < 0.01) after seven days of wash out, and the acute administration with the highest dose of SEP-856 significantly reversed this phenotype (*p* < 0.01 vs. scPCP). On the other end, the lowest dose of SEP-856 (1 mg/kg) did not improve the impaired performance in scPCP-treated animals (*p* < 0.001 vs. controls), as indicated by the strongly reduced DI.

## 3. Discussion

In this study, we first aimed to provide primary evidence for the ability of SEP-856 to modulate specific brain regions under basal conditions. Moreover, we tested its ability to ameliorate the dysfunctions that might be observed under a situation of altered glutamatergic or dopaminergic functionality, using experimental settings that have been already validated with other antipsychotic drugs. In the first explorative experiment we found that an oral administration of SEP-856, given in a range of doses whose antipsychotic ability has been already demonstrated [7], exerted a primary modulatory activity on the prefrontal cortex (PFC) in healthy animals under resting conditions. Furthermore, cortical excitation seemed to be involved in preventing functional and molecular alterations observed in schizophrenia-related dysfunctions. 

We found that an acute treatment with SEP-856 produced a marked elevation of a number of activity dependent genes (IEGs), such as *Arc*, *c-Fos*, *Zif268/Egr1* and *Npas4*, under basal conditions in the prefrontal cortex, and much less in other structures. In line with our data, IEGs expression throughout the brain in response to antipsychotic challenge has been extensively investigated showing region-specific modulations of IEGs expression after antipsychotic treatments [26]. Specifically, atypical antipsychotics such as clozapine induced limbic and prefrontal cortical IEGs expression, whereas a substantial induction in the striatum was found following haloperidol administration [27]. IEGs expression analysis first specifies if a brain area is activated by a certain compound and then may provide information on downstream responses and mechanisms of action that may set the picture for long-term changes set in motion by prolonged drug administration [26,28,29].

In this respect, the PFC appeared to be the brain region showing the most significant changes in response to the acute administration of SEP-856. The modulation of the PFC is shared by different antipsychotic drugs that, after acute or chronic administration, may promote the regulation of IEGs transcription [26,29,30] as well as the modulation of neurotransmitter release [3,31]. The effects of SEP-856 in PFC may depend upon 5-HT_1 A_ activation. Indeed, previous studies have shown an upregulation of different IEGs, including *Arc*, *c-Fos* and *Zif268/Egr1*, within the PFC following administration of the selective serotonin 5-HT_1 A_ receptor agonist (+)-8-hydroxy-2-(di-*n*-propylamino)tetralin ((+)-8-OH-DPAT) [32]. Although administration of the TAAR1 agonist RO5166017 does not modulate c-Fos expression within the PFC [33], we cannot exclude a contribution of TAAR1 agonism in the regulation of other IEGs. The PFC receives serotonergic innervation from the median and dorsal raphe nuclei (DRN) and sends glutamatergic projections to these raphe nuclei [34]. Accordingly, the activation of presynaptic serotonin 5-HT_1 A_ receptors in the DRN autoinhibits the serotonergic firing rate which, in turn, can also be regulated through the glutamatergic projections descending from the PFC [8]. In agreement with this, a previous study on SEP-856 reported that the compound given at the dose of 1 mg/kg could partially suppress firing of the DRN and that this inhibition seemed to still be present 1 h after drug administration [7]. On the contrary, higher doses of SEP-856 (2 and 5 mg/kg) showed an almost complete abolishment of firing rates that were restored to baseline levels 30 min after administration [7]. This evidence might suggest that the lower doses of SEP-856 induce a lower and prolonged inhibitory activity on serotonergic tone, while the highest doses determine a stronger and more immediate effect. Furthermore, although Dedic and colleagues showed that binding of the compound to serotonin 5-HT_1 A_ receptors is dose dependent [7], the modulatory activity of SEP-856 on IEGs expression was already engaged at the lowest dose (1 mg/kg). Whether that might be due to both receptor components of SEP-856 remains to be further evaluated. Indeed, TAAR1 receptors, interacting with both serotonin 5-HT_1 A_ and dopamine D_2_ receptors, are considered important modulators of monoamine transmission [11]. In support of this, the application of TAAR1 agonists displays a strong inhibition in the firing rates of serotoninergic and dopaminergic neurons [11].

Additionally, SEP-856 appears to mostly modulate the ventral part of the hippocampus, as compared to its dorsal counterpart. The hippocampus has an important role in the pathophysiology of schizophrenia since dysfunctions of the human anterior hippocampus, equivalent to the ventral hippocampus in rodents, have been well documented [35,36]. The anterior/ventral hippocampal subregion modulates dopaminergic activity in the VTA and may, therefore, contribute to the hyperdopaminergic state associated with positive symptoms of schizophrenia [37]. Moreover, it has been reported that SEP-856 binding to serotonin 5-HT_1 A_ receptors in the dorsal hippocampus is lower compared to other brain regions [7].

We also investigated the striatum, which represents an important hub for dopaminergic innervation [38]. The activation of this brain area measured by IEG expression has been observed with typical and atypical antipsychotics as haloperidol and lurasidone, whereas no modulation of *Arc*, *c-Fos* and *Zif268/Egr1* has been observed following administration of the selective serotonin 5-HT_1 A_ receptor agonist (+)-8-OH-DPAT [32]. This evidence is in line with our data, since we did not find any significant change of IEGs expression within the striatum after acute SEP-856 administration. 

We showed that an acute pretreatment with SEP-856 was able to counteract the changes produced by phencyclidine (PCP), a valuable tool to induce a hypo-functionality of the glutamatergic system relevant for schizophrenia [39]. Although we used male animals in the first explorative study, we decided to use female rats for the locomotor activity evaluation. Indeed, rats display sex-dependent effects in behavioural responses to psychostimulants, with females being more sensitive to an acute stimulation of amphetamine or phencyclidine [13,14]. As expected, acute PCP produced a significant locomotor hyperactivity that was prevented by the acute pretreatment with SEP-856 at all doses tested. This finding supports the effectiveness of SEP-856 in alleviating the positive manifestations of schizophrenia, and perhaps other effects induced by a hypo-glutamatergic pathology, as already suggested by Dedic and colleagues, who reported a dose-dependent effect of SEP-856 in counteracting the increase of locomotor activity following PCP administration in male mice [7].

Along with this abnormal behaviour, we found molecular perturbation induced by PCP in the PFC. Indeed, the acute administration of the psychostimulant produced an overall increase of IEGs expression, as already reported [40], suggesting a general activation of this brain region. In this regard, the acute pretreatment with SEP-856, at all doses, effectively prevented the upregulation of *Arc* mRNA levels following acute PCP treatment. Furthermore, similar to what we observed after SEP-856 administration, PCP-induced *Arc* overexpression within the PFC was inhibited by pretreatment with clozapine, olanzapine and risperidone [41]. Considering that *Arc* is a downstream effector of the glutamatergic system [42], this result suggests that SEP-856 may promptly act on this pathway, normalizing the alterations of downstream mechanisms in response to an NMDA blockade. Strengthening these observations, we found a significant positive correlation between the total locomotor activity and *Arc* mRNA levels. Indeed, the hyperactivity induced by PCP was correlated with an increased expression of *Arc*, suggesting that the strong activation of cortical regions may reflect the behavioural response to PCP. Moreover, PCP induced a similar activation of *c-Fos*, as previously reported [43,44], although only the highest dose of SEP-856 was able to counteract this effect, suggesting that significant dose differences may exist in the ability of SEP-856 to modulate this pathway under a pathological condition. Taken together, our data suggest that, in a condition of glutamatergic perturbation [17], SEP-856 administration may prevent behavioural alterations by modulating prefrontal cortex functionality. Strengthening these findings and supporting the specific involvement of cortical excitation in preventing the observed alterations, neither the hippocampus nor the striatum showed a similar pattern of changes (data not shown).

Interestingly acute pretreatment with SEP-856, at all tested doses, was not able to prevent the robust increase in locomotor activity observed in rats treated with amphetamine (AMPH) to mimic the hyperdopaminergia found in schizophrenia [13,45]. The lack of effect of SEP-856 in this paradigm may be due to its low affinity for dopaminergic D_2_ receptors [7], which is not sufficient to counteract the strong behavioural activation following AMPH injection [16]. While Dedic and collaborators extensively studied the functional efficacy of the compound after acute PCP administration [7], this is the first work that investigates the antipsychotic-like profile of SEP-856 under AMPH-driven hyperdopaminergic conditions. 

Although SEP-856 did not prevent the strong hyperactivity induced by AMPH, we questioned whether IEGs may be differentially modulated by the compound under an altered dopaminergic condition. Similar to our findings following PCP administration, AMPH strongly increased *Arc* mRNA expression. This result is in agreement with previous studies showing an enhancement of this IEG expression in cortical regions following AMPH administration [46,47]. Interestingly, the acute pretreatment with the highest dose of SEP-856 (10 mg/kg) was able to prevent the robust AMPH-induced effects on *Arc* mRNA levels. While *Arc* may represent a common downstream marker for the activation of PFC in response to systemic administration of PCP and AMPH, the pathways that may drive such effects show differential sensitivity to SEP-856, which is active at all doses in a hypoglutamatergic condition, as compared to the selective effect of the highest dose in a hyperdopaminergic situation. 

Although these data seem be in contrast with what observed in the explorative study, it is feasible that SEP-856 pretreatment is able to activate the PFC at basal conditions, which then promptly responds to the stimulant challenges by counteracting the detrimental effects. However, it should be mentioned that different strains and sex were used, not allowing a direct and precise comparison between the observed effects. 

Last, the highest dose of SEP-856 fully improved the cognitive deficits induced by subchronic treatment with PCP observed in the NOR test, providing predictive validity for improving some aspects of the cognitive dysfunction that represent a key domain of schizophrenia, and unmet clinical need, such as visual learning and memory. We chose a very short retention interval since we aimed to evaluate short-term memory, which is a PFC-dependent task [48]. The effect of acute SEP-856 on subchronic PCP-induced cognitive deficit was comparable to effects shown by atypical antipsychotics, such as clozapine, risperidone and lurasidone [25,49], whereas classical drugs, such as haloperidol, appear to be ineffective [25].

Altogether, these studies were aimed at extending the knowledge on SEP-856 mechanisms of action, providing evidence on its molecular profile within the brain under basal conditions as well as in a pathological situation. However, it should be mentioned that the design of our experiments allowed us to report only a single time point of IEGs induction by drug treatment. Further studies should investigate the effects of a chronic administration of the compound and particularly in animal models that reproduce specific etiological mechanisms of schizophrenia [50]. Interestingly, it has been recently demonstrated that chronic treatment with the selective TAAR1 partial agonist RO5263397 ameliorated chronic stress-induced changes in cognitive function [51], suggesting that this receptor mechanism in SEP-856 may produce long-term changes aimed at correcting the dysfunction of key psychopathologic domains. Moreover, although both male and female animals were employed, we could not evaluate sexual dimorphism in SEP-856 actions, due to the different strains and timing needed to achieve the various goals of the study. 

Nevertheless, we believe that our approach still provides useful information regarding the acute effects of SEP-856, as well as its anatomical specificity. The preferential action of SEP-856 on cortical regions could contribute to the clinical effects reported in phase II clinical trials with schizophrenic patients. Indeed, the PFC is mainly involved in executive functions and working memory, which are altered in schizophrenia. Furthermore, it is enriched in neurons expressing serotonin 5-HT_1 A_ receptors, which may support the primary action of SEP-856 in this brain area. Lastly, while we did not compare the effects of SEP-856 to those of compounds that are selective for TAAR1 or 5-HT_1 A_, the aim of our study was to specifically evaluate SEP-856 as a whole under conditions that may be encountered in the clinical setting.

## 4. Materials and Methods

### 4.1. Animals

#### 4.1.1. Acute SEP-363856 Administration

This experiment was performed at the University of Milan. Adult male Sprague Dawley rats (310–370 g) (*n* = 40, 10 animals per group; Charles River Laboratories, Italy) were housed in groups of three in plastic cages containing paper and sawdust (45 cm × 28 cm × 20 cm), under standard laboratory conditions (standard rat chow and water available ad libitum) on a 12-h light/dark cycle (lights on at 07:00) in constant temperature (22 ± 2 °C) and humidity (50 ± 5%) conditions.

All animal experiments were conducted according to the authorization from the Health Ministry *n*. 1252/2015-PR, in full accordance with the Italian legislation on animal experimentation (Decreto Legislativo 26/2014) and adherent to EU recommendation (Directive 2010/63/EU). All efforts were made to minimize animal suffering and to reduce the total number of animals used, while maintaining statistically valid group numbers.

No pre-established inclusion/exclusion criteria were used for the subsequent molecular analyses. All samples were processed and analysed by investigators blind to the treatment condition. All procedures were conducted in the morning (at 11:00 a.m. ± 2).

#### 4.1.2. Phenciclidine and D-Amphetamine Injections

These studies were performed by b-neuro at the University of Manchester. Female Lister Hooded (LH) rats (190–220 g) (*n* = 50 per experiment, 10 animals per group; Charles River Laboratories, UK), were housed in groups of five in individually ventilated, two-tiered plastic cages (38 cm × 59 cm × 24 cm, GR1800 Double-Decker Cage, Tecniplast, UK). These cages contained paper sizzle nest, sawdust, and cardboard tunnels (Datesand group, UK). The rats had access to water and standard rat chow (Special Diet Services) ad libitum within the home cage. The environment was kept constant at 20 ± 2 °C, 55 ± 5% humidity, under a 12:12 h light/dark cycle (lights on at 07:00). 

All procedures used in these experiments were conducted in accordance with the Animals Scientific Procedures Act (London, UK, 1986) and were approved by the University of Manchester AWERB (Animal Welfare and Ethical Review Body).

No pre-established inclusion/exclusion criteria were used for the subsequent molecular analyses. All samples were processed and analysed by investigators blind to the housing conditions. All procedures were conducted in the morning (at 11:00 a.m. ± 2).

### 4.2. Pharmacological Treatment

#### 4.2.1. Acute SEP-363856 Administration

After two weeks of adaptation to laboratory conditions, animals received an oral administration of SEP-363856 (SEP-856; 1, 3 or 10 mg/kg) or vehicle (50 mm acetate buffer at pH 5.4). These doses were chosen based on a previous characterization of the compound. Indeed, within these dose ranges, the compound showed antipsychotic activity [7]. SEP-856 was kindly provided by Sunovion Pharmaceuticals. Sunovion discovered SEP-363856 in collaboration with PsychoGenics based in part on a mechanism-independent approach using the in vivo phenotypic SmartCube^®^ platform and associated artificial intelligence algorithms. SEP-856 was prepared by suspending the drug in 50 mM acetate buffer at pH 5.4. The chemical structure of SEP-856 is reported in [7]. Vehicle and drug were administered via oral gavage in the amount of 1 mL/kg body weight. Sixty minutes after drug administration, all the animals were sacrificed by decapitation and the brain regions of interest (hippocampus -dorsal and ventral- prefrontal cortex and striatum) were immediately dissected, frozen on dry ice and stored at −80 °C for the molecular analyses. The 60 min time period was chosen based on previous studies, and since immediate-early genes are rapidly induced [7,52]. 

#### 4.2.2. Acute Phencyclidine and D-Amphetamine Treatments

After two weeks of adaptation to laboratory conditions, animals received an oral administration of SEP-856 (1, 3 or 10 mg/kg) or vehicle (50 mM acetate buffer at pH 5.4). Following the 60 min pretreatment time, rats received a second injection of vehicle (0.9% saline), phencyclidine (PCP, 2.0 mg/kg, i.p., Sigma, Welwyn Garden City, UK, LOT126 M4075 V) or *d*-amphetamine (AMPH, 0.1 mg/kg, i.p. Sigma, Gillingham, Dorset, UK). Ninety minutes after the second injections, following locomotor activity assessment, animals were sacrificed by decapitation (Figure 8A). The brains were removed and the prefrontal cortex was collected. All samples were then rapidly frozen on dry ice and stored at −80 °C for the molecular analyses.

#### 4.2.3. Subchronic Phencyclidine Treatment

After two weeks of adaptation to laboratory conditions, animals were treated with PCP (2.0 mg/kg, i.p., Sigma, Gillingham, Dorset, UK, LOT 126M4075 V) or vehicle (0.9% saline) twice daily for seven days. Following a seven-day drug-free period, rats received an oral administration of SEP-856 (1 or 10 mg/kg) or vehicle (50 mM acetate buffer at pH 5.4). Sixty minutes later, animals performed the novel object recognition (NOR) test (Figure 8B). We decided to test only two doses of SEP-856 in order to evaluate the effects of the two extreme dosages, as well as to comply with the 3Rs principle by reducing the number of animals used.

The doses of PCP (2.0 mg/kg) and AMPH (0.1 mg/kg) used in these studies were based on previous in-house studies in subchronic PCP-treated female Lister Hooded rats. Results demonstrated an enhanced sensitivity to the locomotor stimulant effects of a subsequent acute challenge with PCP [53] or AMPH (Munni, unpublished findings).

### 4.3. Behavioural Testing

#### 4.3.1. Locomotor Activity Evaluation

A photo beam activity system (PAS version 2.0) from San Diego Instruments (SDI) was used to record locomotor activity in acute PCP and AMPH experiments. The system works by recording beam interruptions across the x & y axis (4 × 8 beams) of the lower frame (4 cm from floor) and across the y axis (eight beams) on the top frame (20 cm from floor). Central and peripheral activity was recorded separately. The automated recording eliminates the subjective influence of experimenters in the study and also gives the experimenter the ability to measure locomotor activity to a high level of detail. The testing box is rectangular and made of clear Plexiglas (H: 21 cm, W: 30 cm, L: 52 cm).

##### Habituation Phase

Rats were allowed to habituate to the test box and the behavioural test room for 1 h per day for three days prior to testing. Each test box was covered with a layer of bedding material. On the day of testing, rats were further habituated to the test room for 30 min.

##### Behavioural Testing

Following the 30 min habituation period to the test room, rats were treated with vehicle or SEP-856 (1, 3 & 10 mg/kg, p.o.) and immediately placed into the same test box they were in during habituation. Following the 60 min pretreatment time, rats received a second injection of vehicle, PCP or AMPH and were returned to the test box for a further 90 min evaluation. Testing was carried out in the light phase.

##### Locomotor Activity

The automatic PAS system recorded beam breaks every five minutes for a total period of 150 min, and the central and peripheral count was recorded. 

The protocol used was based on previous locomotor studies in our laboratory [53] with minor modifications.

#### 4.3.2. Novel Object Recognition Test

Recognition memory was evaluated in the subchronic PCP (scPCP) experiment using the novel object recognition (NOR) test [25]. The apparatus consisted of an open black Plexiglas box (L: 52 cm, W: 52 cm, H: 31 cm), positioned 27 cm above the floor. 

##### Habituation Phase

Rats were habituated to the empty test box and the behavioural test room environment for 1 h in their cage groups. The day after, prior to behavioural testing, rats were given a further 3 min habituation.

##### Behavioural Testing

The test consisted in two 3 min-trials separated by 1 min of inter-trial interval in the home cage. During the first trial, rats were individually placed into the same test box they were in during habituation, allowed to explore two identical objects for 3 min. During the 1-min intertrial interval that the animals spent in their home cage, the two identical objects were replaced with a familiar object, corresponding to the objects used during the previous trial, and a novel object different in colour, shape and material. The familiar object used was a duplicate of the object presented during the first trial to avoid any olfactory bias. 

During the second trial, rats were allowed to explore the familiar and novel objects in a 3 min retention trial. 

Both the trials were recorded on video for the subsequent blind scoring. The object exploration was defined by animal licking, sniffing or touching the objects with the forepaws. 

The exploration time of each object in each trial was measured and the discrimination index (DI) was calculated as shown below:(1)$$DI=\frac{time\;spent\;exploring\;the\;novel\;object-time\;exploring\;the\;familiar\;object}{total\;time\;spent\;in\;exploring\;the\;objects}$$

### 4.4. RNA Preparation and Quantitative Real-Time PCR Analyses

For gene expression analysis, total RNA was isolated by single step guanidinium isothiocyanate/phenol extraction using PureZol RNA isolation reagent (Bio-Rad Laboratories) according to the manufacturer’s instructions. Following total RNA extraction, an aliquot was treated with DNase (DNase I, RNase-free; ThermoScientific) to avoid DNA contamination, and the samples were subsequently processed for reverse-transcriptase real-time polymerase chain reaction (qRT-PCR), as previously described [54], to assess mRNA levels of: Activity-Regulated Cytoskeleton-Associated Protein (*Arc/Agr3.1*), Cellular Oncogene C-Fos (*c-Fos*), Zinc Finger Binding Protein clone 268 (*Zif268/Egr1*) and Neuronal PAS Domain Protein 4 (*Npas4*).

In detail, 10 ng of purified RNA were analysed by TaqMan qRT-PCR instrument (CFX384 real-time system, Bio-Rad Laboratories) using the iScript one-step RT-PCR kit for probes (Bio-Rad Laboratories). Thermal cycling was initiated with an incubation at 50 °C for 10 min (RNA retrotranscription), followed by a 5-min period at 95 °C (TaqMan polymerase activation). After this initial step, 39 cycles of PCR were performed. Each PCR cycle consisted of heating the samples for 10 s at 95 °C to enable the melting process and then for 30 s at 60 °C for the annealing and extension reactions.

Primers and probes sequences were purchased from Eurofins Genomics and are summarized in Table 1. Samples were run in triplicate as multiplexed reactions with a normalizing internal control (*ß-actin*). Relative target gene expression was calculated according to the 2-D(D(Ct)) method. To simplify graphical representations, the obtained data were expressed as percentage versus the control group, which was set at 100%.

### 4.5. Statistical Analyses

All the analyses were carried out in individual animals (independent determinations). The outliers were determined using IBM SPSS Statistics 23 and accordingly removed from the analyses. Specifically, the molecular effects of an acute administration of SEP-856, as well as behavioural and molecular effects of PCP and AMPH were analysed by one-way analysis of variance (ANOVA). Repeated measures ANOVA was used to evaluate the effects of drug treatment in the comprehensive analysis of locomotor activity over the total 150 min assessment. Student’s t test was used to analyse the time spent exploring the two identical objects in the acquisition trial, as well as the familiar and novel object in the retention trial of the NOR. When appropriate, further differences were analysed by post hoc comparison. In detail, behavioural and molecular data were further analysed by Tukey post hoc comparisons using IBM SPSS Statistics 23. Significance for all tests was assumed for *p* < 0.05.

## 5. Conclusions

In summary, we report that SEP-856 may exert a neuromodulatory activity by promoting the expression of plasticity-related genes under basal conditions, with a main effect in the prefrontal cortex. Furthermore, we demonstrate the ability of this novel compound in preventing the abnormal behaviour and in restoring activity-regulated gene expression in a hypoglutamatergic condition. Our study provides new insights into the activity of the novel putative antipsychotic compound SEP-856, suggesting that it could contribute to neuroadaptive changes of relevance for the treatment of schizophrenia. To date, this is the first evidence for the ability of SEP-856 to improve a certain aspect of cognition in schizophrenia.

## Figures and Tables

**Figure 1 ijms-22-04119-f001:**
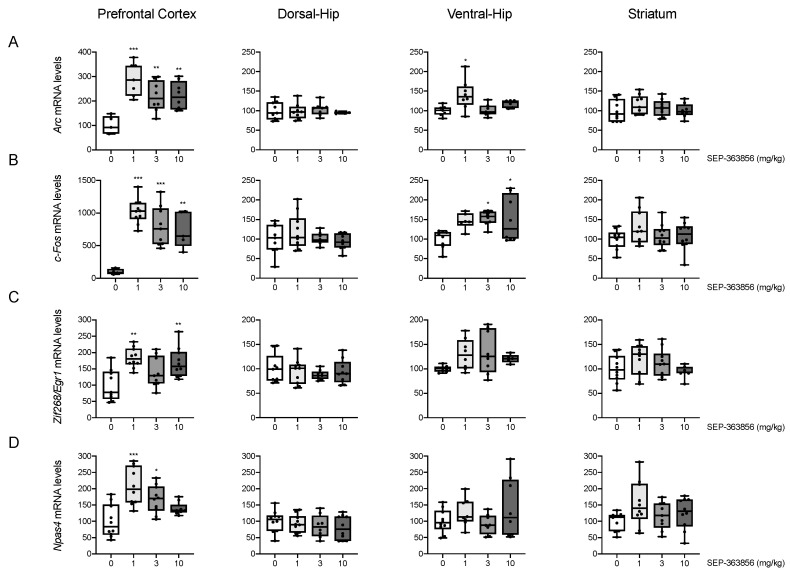
Modulation of IEGs expression following acute SEP-856 administration. The mRNA levels of *Arc* (**A**), *c-Fos* (**B**)*, Zif268/Egr1* (**C**) and *Npas4* (**D**) were measured in different brain regions of rats treated with vehicle or SEP-856 (1, 3 or 10 mg/kg). The data, expressed as percentage of vehicle-treated rats set at 100%, are represented as box-and-whisker plots of at least four independent determinations. For each box, the box boundaries indicate the first and third quartiles, the middle line specifies the median and the whiskers represent the lowest and highest values. * *p* < 0.05, ** *p* < 0.01, *** *p* < 0.001 vs. vehicle-treated rats (one-way ANOVA followed by Tukey post-hoc comparison).

**Figure 2 ijms-22-04119-f002:**
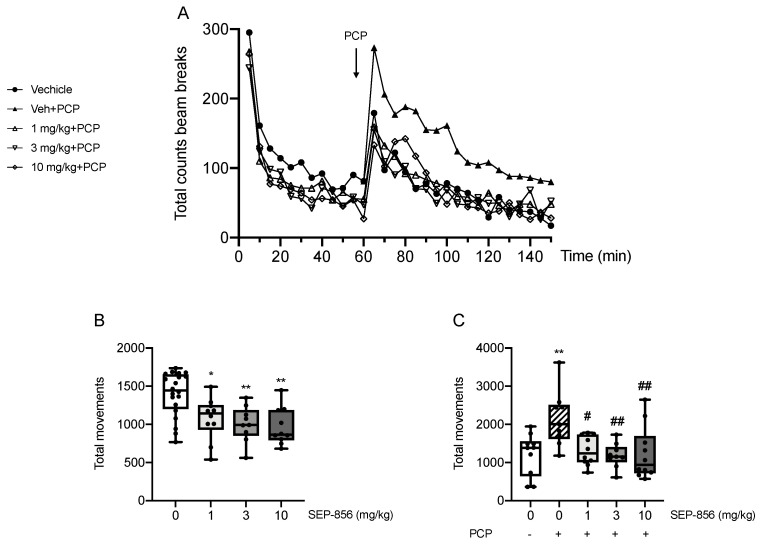
Analysis of total locomotor activity. Total locomotor activity counts over 150 min (**A**). The first 60 min show locomotor activity after treatment with SEP-856 (1, 3 & 10 mg/kg, p.o.) or vehicle. At 60 min rats received acute injection of PCP (2 mg/kg, i.p.) or vehicle, as indicated by the arrow, and locomotor activity was measured for further 90 min. The total activity was measured over the first 60 min (**B**) pretreatment with different doses of SEP-856 (1, 3, 10 mg/kg) and over the 90 min period (**C**) following PCP administration. Data are represented as box-and-whisker plots (**B** and **C**) of at least nine independent determinations. For each box, the box boundaries indicate the first and third quartiles, the middle line specifies the median and the whiskers represent the lowest and highest values. * *p* < 0.05, ** *p* < 0.01 vs. vehicle-treated rats; # *p* < 0.05, ## *p* < 0.01 vs. PCP-treated rats (one-way ANOVA followed by Tukey post-hoc comparison).

**Figure 3 ijms-22-04119-f003:**
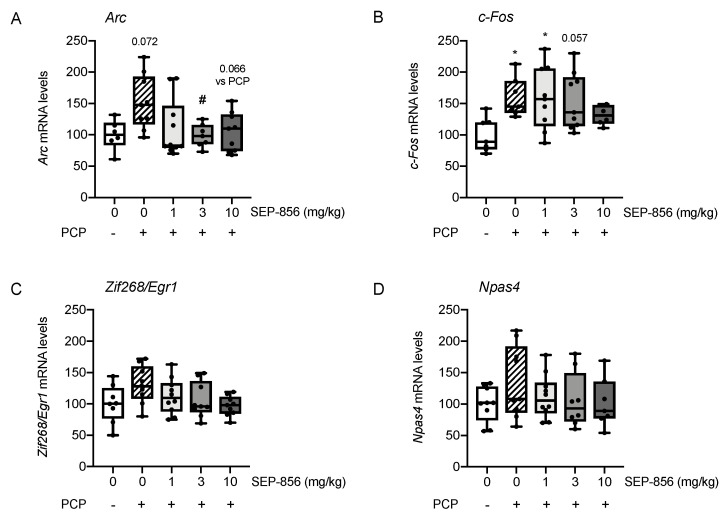
Modulatory activity of acute SEP-856 treatment on IEG changes after acute PCP administration. The mRNA levels of *Arc* (**A**), *c-Fos* (**B**), *Zif268/Egr1* (**C**) and *Npas4* (**D**) were measured in the prefrontal cortex of rats pretreated with vehicle or SEP-856 (1, 3 or 10 mg/kg) following the acute injection of PCP (2 mg/kg). The data, expressed as percentage of vehicle-treated rats set at 100%, are represented as box-and-whisker plots of at least six independent determinations. For each box, the box boundaries indicate the first and third quartiles, the middle line specifies the median and the whiskers represent the lowest and highest values. * *p* < 0.05 vs. vehicle-treated rats; # *p* < 0.05 vs. PCP-treated rats (one-way ANOVA followed by Tukey post-hoc comparison).

**Figure 4 ijms-22-04119-f004:**
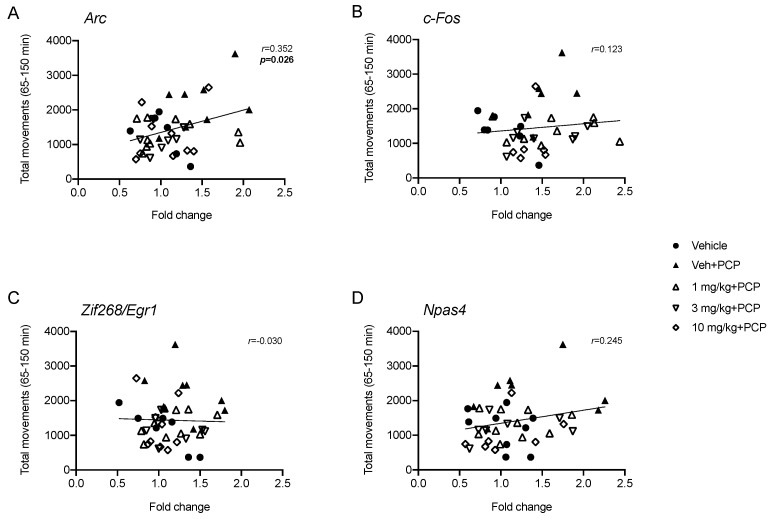
Pearson product–moment correlation (*r*) between the total counts measured in the second part of the test (65–150 min) and gene expression of *Arc* (**A**), *c-Fos* (**B**), *Zif268/Egr1* (**C**) and *Npas4* (**D**) following PCP injection. The statistical significance was assumed with *p* < 0.05.

**Figure 5 ijms-22-04119-f005:**
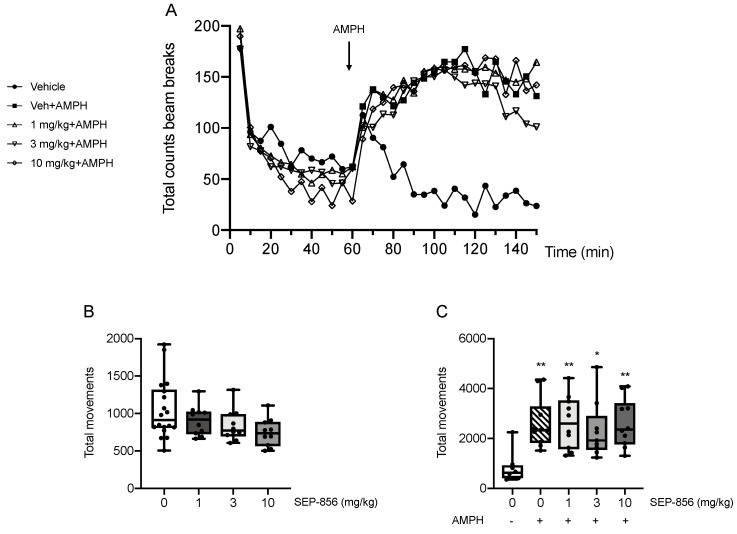
Analysis of total locomotor activity. Total locomotor activity counts over 150 min (**A**). The first 60 min show locomotor activity after treatment with SEP-856 (1, 3 & 10 mg/kg, p.o.) or vehicle. At 60 min rats received acute injection of AMPH (0.1 mg/kg, i.p.) or vehicle, as indicated by the arrow, and locomotor activity was measured for further 90 min. The total activity was measured over the first 60 min (**B**) pretreatment with different doses of SEP-856 (1, 3, 10 mg/kg) and over the 90 min period (**C**) following AMPH administration. Data are represented as box-and-whisker plots of at least nine independent determinations (**B** and **C)**. For each box, the box boundaries indicate the first and third quartiles, the middle line specifies the median and the whiskers represent the lowest and highest values. * *p* < 0.05, ** *p* < 0.01 vs. vehicle-treated rats (one-way ANOVA followed by Tukey post hoc comparison).

**Figure 6 ijms-22-04119-f006:**
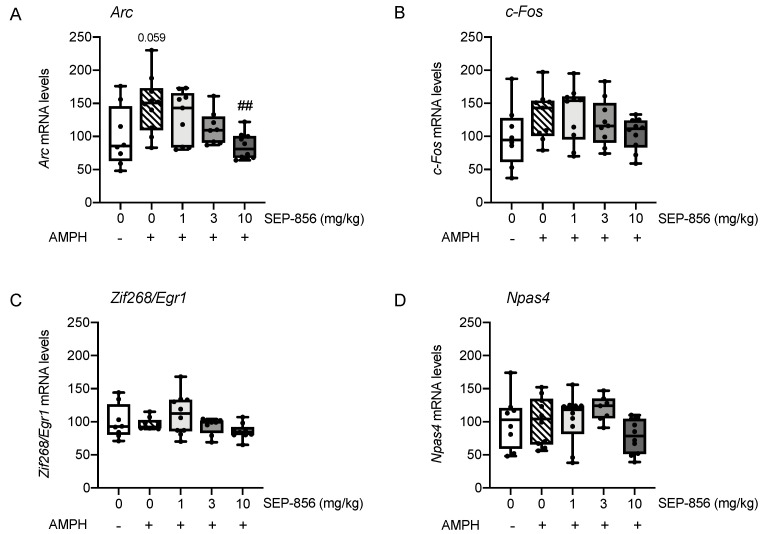
Modulatory activity of acute SEP-856 treatment on IEG changes after AMPH administration. The mRNA levels of Arc (**A**), c-Fos (**B**), Zif268/Egr1 (**C**) and Npas4 (**D**) were measured in the prefrontal cortex of rats pretreated with vehicle or SEP-856 (1, 3 or 10 mg/kg) following the acute injection of AMPH (0.1 mg/kg). The data, expressed as percentage of vehicle-treated rats set at 100%, are represented as box-and-whisker plots of at least seven independent determinations. For each box, the box boundaries indicate the first and third quartiles, the middle line specifies the median and the whiskers represent the lowest and highest values. ## *p* < 0.01 vs. AMPH-treated rats (one-way ANOVA followed by Tukey post hoc comparison).

**Figure 7 ijms-22-04119-f007:**
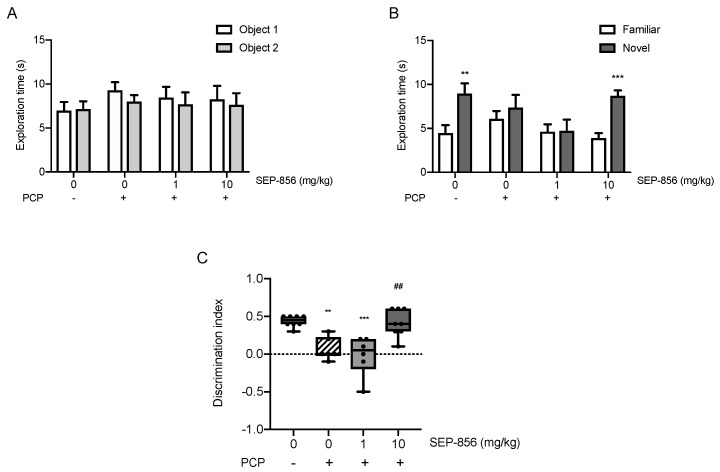
Effects of acute SEP-856 administration on cognitive impairment following the scPCP regimen. Time spent exploring the two identical objects in the acquisition trial (**A**). Time spent exploring the familiar and the novel object in the retention trial (**B**). ** *p* < 0.01, *** *p* < 0.001 significant differences in the time spent exploring the novel object as compared to the familiar one. Data are expressed as mean ± SEM of at least six independent determinations. NOR performance is expressed as discrimination index (**C**), representing the difference between time spent exploring novel and familiar objects during the testing phase. The data, expressed as percentage of vehicle-treated rats set at 100%, are represented as box-and-whisker plots of at least seven independent determinations. For each box, the box boundaries indicate the first and third quartiles, the middle line specifies the median and the whiskers represent the lowest and highest values. ** *p* < 0.01, *** *p* < 0.001 vs. vehicle-treated rats; ## *p* < 0.01 vs. scPCP-treated rats (one-way ANOVA followed by Tukey post hoc comparison).

**Figure 8 ijms-22-04119-f008:**
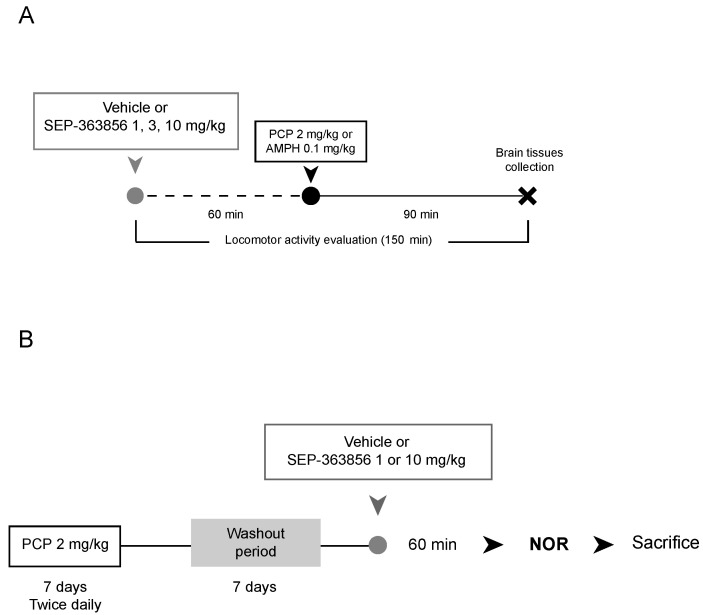
Schematic representation of the experimental paradigms. Acute Phencyclidine and D-Amphetamine Treatments (**A**). Animals received an oral administration of SEP-856 or vehicle. Following the 60 min pretreatment time, rats received a second injection of vehicle, phencyclidine (PCP) or d-amphetamine (AMPH). Ninety minuites after the second injections, following locomotor activity assessment, animals were sacrificed by decapitation (**A**). Subchronic Phencyclidine Treatment (**B**). Animals were treated with phencyclidine (PCP) or vehicle twice daily for seven days. Following a seven-day drug-free period, rats received an oral administration of SEP-856 or vehicle. Sixty minutes later, animals performed the novel object recognition (NOR) test (**B**).

**Table 1 ijms-22-04119-t001:** Sequences of Forward and Reverse primers and probes used in qRT-PCR analyses.

Gene	Forward Primer	Reverse Primer	Probe
*Arc/Agr3.1*	GGTGGGTGGCTCTGAAGAAT	ACTCCACCCAGTTCTTCACC	GATCCAGAACCACATGAATGGG
*C-Fos*	TCCTTACGGACTCCCCAC	CTCCGTTTCTCTTCCTCTTCAG	TGCTCTACTTTGCCCCTTCTGCC
*Zif268/Egr1*	GAGCGAACAACCCTACGAG	GTATAGGTGATGGGAGGCAAC	TCTGAATAACGAGAAGGCGCTGGTG
*Npas4*	TCATTGACCCTGCTGACCAT	AAGCACCAGTTTGTTGCCTG	TGATCGCCTTTTCCGTTGTC
*ß-Actin*	CACTTTCTACAATGAGCTGCG	CTGGATGGCTACGTACATGG	TCTGGGTCATCTTTTCACGGTTGGC

## Data Availability

The data presented in this study are available on request from the corresponding author.

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
