# Peer review of "Towards Novel Treatments for Schizophrenia: Molecular and Behavioural Signatures of the Psychotropic Agent SEP-363856"

_ijms, 2021, doi:10.3390/ijms22084119_

Round 1

Reviewer 1 Report

Many thanks for the invitation to review this paper. Authors have conducted a comprehensive analysis of the effects of the psychotropic agent SEP-363856, with relevance for schizophrenia, on expression of early immediate genes, and behavioral/cognitive parameters of rats. They presented substantial evidence that SEP-363856 is indeed able to modulate the expression of Arc in the prefrontal cortex, and attenuate behavioral and cognitive deficits induced by phencyclidine. The manuscript is very well written, with robust methods for the actual analyses performed. 

A main finding of this study is that SEP-363856 was able to rescue cognitive impairment following phencyclidine regimen on novel object recognition task. This effect was detected only with a high dose of SEP-363856 (10mg/kg), and not with 1mg/kg. However, important results throughout the paper were observed with a dose of 3mg/kg of SEP-363856. Indeed, this was the dose with the highest effect on the rescue of Arc expression following phencyclidine treatment. Therefore, It would substantially improve the manuscript if the authors presented data on the novel object recognition task in conditions of 3mg/kg of SEP-363856 following phencyclidine regimen.

Author Response

Reviewer 1

Many thanks for the invitation to review this paper. Authors have conducted a comprehensive analysis of the effects of the psychotropic agent SEP-363856, with relevance for schizophrenia, on expression of early immediate genes, and behavioral/cognitive parameters of rats. They presented substantial evidence that SEP-363856 is indeed able to modulate the expression of Arc in the prefrontal cortex, and attenuate behavioral and cognitive deficits induced by phencyclidine. The manuscript is very well written, with robust methods for the actual analyses performed.

A main finding of this study is that SEP-363856 was able to rescue cognitive impairment following phencyclidine regimen on novel object recognition task. This effect was detected only with a high dose of SEP-363856 (10mg/kg), and not with 1mg/kg. However, important results throughout the paper were observed with a dose of 3mg/kg of SEP-363856. Indeed, this was the dose with the highest effect on the rescue of Arc expression following phencyclidine treatment. Therefore, It would substantially improve the manuscript if the authors presented data on the novel object recognition task in conditions of 3mg/kg of SEP-363856 following phencyclidine regimen.

Response

We thank the reviewer for the suggestion, and we agree that it would be of great interest to evaluate also the effects of SEP-856 at the 3mg/kg dose on the cognitive performance. However, in order to comply with the 3Rs principles for reducing the number of animals used, we had limitation in the number of animals to be used and therefore we decided not to include the intermediate dosage in our study (Methods section 4.2.3).

Reviewer 2 Report

This is an interesting study, which provided new insights into the mechanisms of action of SEP363-856.  The manuscript is well written and the methods used are appropriate. 

The following is offered to improve the manuscript:

(1) Is SEP856-induced modulation of PFC selective for this compound or is it shared by other atypical antipsychotics.

(2) What are the difference between this compound and lurasidone.

(3) Any data or speculation on what are the chronic effects of SEP856 on IEGs

(4) The authors should elaborate more on the clinical implications of SEP856-induced modulation of PFC, and how this modulation can contribute to the observed clinical effects reported in Phase 2 trial in schizophrenia patients.

Author Response

Reviewer 2

This is an interesting study, which provided new insights into the mechanisms of action of SEP363-856.  The manuscript is well written and the methods used are appropriate.

The following is offered to improve the manuscript:

  • Is SEP856-induced modulation of PFC selective for this compound or is it shared by other atypical antipsychotics.

Response

We thank the reviewer for raising this issue. Indeed, the modulation of the PFC is shared by different antipsychotic drugs, after acute or chronic administration, promoting the regulation of IEGs transcription [1–3] as well as the modulation of neurotransmitter release [4,5]. Furthermore, similar to what we observed after SEP-856 administration, PCP-induced Arc overexpression within the PFC is inhibited by pretreatment with clozapine, olanzapine and risperidone [6].

Two short sentences have been added in the discussion (line 356 and 419).

  • What are the difference between this compound and lurasidone.

Response

We thank the reviewer for asking this question. Lurasidone is characterized by a multi-receptor profile. It is a full antagonist at dopamine D2 and at serotonin 5-HT2A and 5-HT7 receptors as well as a partial agonist at serotonin 5-HT1A receptor. Hence, based on the receptor profile, the partial agonism at 5-HT1A receptors is the only mechanism shared between lurasidone and SEP-856. Nevertheless, mechanisms downstream from receptor activation highlight similarities and differences. Indeed, following acute administration [1], lurasidone increases Arc expression in the hippocampus, an effect found with the lower dose of SEP-856 in the ventral part of this region. On the contrary, no significant changes in IEG expression are found in the PFC after acute lurasidone, as compared to the large effects found with SEP-856 (present study). Lastly, the modulation of Arc expression in the striatum is only observed with the higher dose of lurasidone, but not with SEP-856. However, we have also demonstrated that chronic treatment with lurasidone is effective in ameliorating molecular and functional changes observed in the PFC of animals exposed to paradigms (prenatal stress or chronic mild stress) that reproduce etiological mechanisms of psychiatric disorders. This may suggest similar benefits when compared to the effects observed for SEP-856 in the PCP model.

While we cite some of these papers, only limited information on this issue is included in the discussion. Indeed, we tried to discuss the profile of SEP-856 in a broader way as compared to antipsychotic drugs in general.

  • Any data or speculation on what are the chronic effects of SEP856 on IEGs

Response

Unfortunately, we do not have data on the effects of chronic SEP-856 treatment. We believe that it will be important to investigate such effects in animal models that reproduce specific pathologic domains that are affected in schizophrenia. However, this will be the goal of future studies. A short sentence has been included in the discussion (line 476).

  • The authors should elaborate more on the clinical implications of SEP856-induced modulation of PFC, and how this modulation can contribute to the observed clinical effects reported in Phase 2 trial in schizophrenia patients.

Response

We thank the reviewer for the suggestion, and we have included a sentence in the discussion (line 487). Unfortunately, the clinical data on SEP-856 are still limited and we are waiting the results of phase III studies. With this respect, imaging studies will be instrumental to provide support for the capability of SEP-856 treatment in modulating cortical function as a key mechanism in its clinical activity.

References

  1. Luoni, A.; Rocha, F.F.; Riva, M.A. Anatomical specificity in the modulation of activity-regulated genes after acute or chronic lurasidone treatment. Prog. Neuro-Psychopharmacology Biol. Psychiatry 2014, 50, 94–101, doi:10.1016/j.pnpbp.2013.12.008.
  2. Luoni, A.; Fumagalli, F.; Racagni, G.; Riva, M.A. Repeated aripiprazole treatment regulates Bdnf, Arc and Npas4 expression under basal condition as well as after an acute swim stress in the rat brain. Pharmacol. Res. 2014, 80, 1–8, doi:10.1016/j.phrs.2013.11.008.
  3. De Bartolomeis, A.; Buonaguro, E.F.; Latte, G.; Rossi, R.; Marmo, F.; Iasevoli, F.; Tomasetti, C. Immediate-early genes modulation by antipsychotics: Translational implications for a putative gateway to drug-induced long-term brain changes. Front. Behav. Neurosci. 2017, 11, 240, doi:10.3389/fnbeh.2017.00240.
  4. Aringhieri, S.; Carli, M.; Kolachalam, S.; Verdesca, V.; Cini, E.; Rossi, M.; McCormick, P.J.; Corsini, G.U.; Maggio, R.; Scarselli, M. Molecular targets of atypical antipsychotics: From mechanism of action to clinical differences. Pharmacol. Ther. 2018, 192, 20–41, doi:10.1016/j.pharmthera.2018.06.012.
  5. Horacek, J.; Bubenikova-Valesova, V.; Kopecek, M.; Palenicek, T.; Dockery, C.; Mohr, P.; Höschl, C. Mechanism of action of atypical antipsychotic drugs and the neurobiology of schizophrenia. CNS Drugs 2006, 20, 389–409, doi:10.2165/00023210-200620050-00004.
  6. Nakahara, T.; Kuroki, T.; Hashimoto, K.; Hondo, H.; Tsutsumi, T.; Motomura, K.; Ueki, H.; Hirano, M.; Uchimura, H. Effect of atypical antipsychotics on phencyclidine-induced expression of arc in rat brain. Neuroreport 2000, 11, 551–555, doi:10.1097/00001756-200002280-00025.
